# Arabic Toxic Tweet Classification: Leveraging the AraBERT Model

**Amr Mohamed El Koshiry [1,2], Entesar Hamed I. Eliwa [3,4], Tarek Abd El-Hafeez [4,5,*] and Ahmed Omar [4,*]**

1 Department of Curricula and Teaching Methods, College of Education, King Faisal University, P.O. Box 400, Al-Ahsa 31982, Saudi Arabia; aalkoshiry@kfu.edu.sa or al_koshiry@mu.edu.eg
2 Faculty of Specific Education, Minia University, Minia 61519, Egypt
3 Department of Mathematics and Statistics, College of Science, King Faisal University, P.O. Box 400, Al-Ahsa 31982, Saudi Arabia; eheliwa@kfu.edu.sa or entesar.eliwa@mu.edu.eg
4 Department of Computer Science, Faculty of Science, Minia University, Minia 61519, Egypt
5 Computer Science Unit, Deraya University, Minia 61765, Egypt
* Correspondence: tarek@mu.edu.eg (T.A.E.-H.); ahmed.omar@mu.edu.eg (A.O.)

**Abstract:** Social media platforms have become the primary means of communication and information sharing, facilitating interactive exchanges among users. Unfortunately, these platforms also witness the dissemination of inappropriate and toxic content, including hate speech and insults. While significant efforts have been made to classify toxic content in the English language, the same level of attention has not been given to Arabic texts. This study addresses this gap by constructing a standardized Arabic dataset specifically designed for toxic tweet classification. The dataset is annotated automatically using Google's Perspective API and the expertise of three native Arabic speakers and linguists. To evaluate the performance of different models, we conduct a series of experiments using seven models: long short-term memory (LSTM), bidirectional LSTM, a convolutional neural network, a gated recurrent unit (GRU), bidirectional GRU, multilingual bidirectional encoder representations from transformers, and AraBERT. Additionally, we employ word embedding techniques. Our experimental findings demonstrate that the fine-tuned AraBERT model surpasses the performance of other models, achieving an impressive accuracy of 0.9960. Notably, this accuracy value outperforms similar approaches reported in recent literature. This study represents a significant advancement in Arabic toxic tweet classification, shedding light on the importance of addressing toxicity in social media platforms while considering diverse languages and cultures.

**Keywords:** Arabic toxic; toxic classification; Arabic NLP; BERT

## 1. Introduction

Online social media platforms are the technological advancement of the 21st century with the greatest cultural impact. Among their many advantages is the broad diffusion of content across countries and the facilitation of engagement and exchanges almost devoid of physical restrictions, except in infrastructure [1].

Social media enables users from various walks of life to connect with like-minded people and build networks that benefit both parties. These users come from different backgrounds of religion, ethnicity, politics, and ethics [2]. In social media, people can express themselves, their cultures, and their viewpoints and converse with others. It stands to reason that viewpoint disagreements could lead to discussions. These arguments frequently take a negative turn, with some leading to online arguments in which one side may use abusive language or toxic comments [3].

Toxic content includes threats, offense, vulgarity, insults, and identity-based hatred, and it clearly presents the risk of online abuse and harassment. Toxic comments set off a chain reaction, with someone writing something nasty in the comment box, prompting the targeted individual to leave the conversation or stop expressing an alternative opinion.

Others then contribute to this poison by using more harsh language or challenging the targeted people in their language, fostering unhealthy and unjust dialogues [2].

Words, phrases, or statements that imply or express contempt for other people are referred to as toxic comments [4]. These comments result to feelings of degradation or insult. Communities are seeing an increase in toxic content, regardless of whether the concerned parties know one another. Social media sites have sought to combat damaging content, but their efforts have mostly been unsuccessful and have had unexpected consequences. The commercial interests or political and regulatory reasons of these platforms may skew the efficacy of their moderation [2].

Toxicity classification methods include biases that are pervasive in society from society-generated training data and repeat these biases in classification outcomes (e.g., frequently linking attacked identity groups such as "Black" and "Muslim" with toxicity even in harmless situations) [4].

The majority of harmful classification efforts have been focused on the English language. Attempts to do the same for the Arabic language, which faces numerous challenges, have been limited. Arabic is the fourth most popular online language. As a result, Arabic material has increased rapidly in recent years. Arabic material has surpassed 3% of total online content and presently ranks seventh in terms of content on the internet. With over 226 million Arabic internet users globally, the Arabic language is rich in material [4].

One-third of the Arabic content available online has low quality and is produced by social network users [5]. Arabic datasets have been used in a few studies and receive little support. For this reason, Arabic text analysis and classification have become increasingly necessary.

The main contributions of this work are as follows:

1. We curated a publicly accessible dataset consisting of 31,836 Arabic tweets, meticulously annotated as toxic or non-toxic. This novel dataset is a standardized resource crafted for the explicit purpose of toxic tweet classification, aiming to fill the gap in toxic content analysis for Arabic texts.
2. Automated Annotation and Expertise: The dataset is annotated using Google's Perspective API combined with the expertise of three native Arabic speakers and linguists, ensuring a comprehensive and accurate labeling process.
3. Model Evaluation: Seven different models, including LSTM, GRU, CNN, and multilingual BERT, are evaluated to determine their performance in toxic tweet classification. The fine-tuned AraBERT model emerges as the top performer, surpassing other models with an impressive accuracy of 0.9960.
4. Superior Performance: The accuracy achieved by the AraBERT model outperforms similar approaches reported in the recent literature, highlighting its effectiveness in accurately identifying toxic content in Arabic tweets.
5. Advancement in Arabic Toxic Tweet Classification: This study signifies a significant advancement in Arabic toxic tweet classification, shedding light on the importance of addressing toxicity in social media platforms while considering the diverse languages and cultures involved.

The remainder of this paper is organized as follows: Section 2 reviews the most recent related work on toxic and hate speech detection; Section 3 elucidates the models utilized in the classification process; Section 4 presents the methodology encompassing dataset construction, annotation, preprocessing steps, and the classification model; Section 5 delves into the experimental results; Sections 6 and 7 represent the discussions and limitations; and Section 8 concludes the study and outlines potential future work.

## 2. Related Work

Toxicity detection has gained popularity as a research topic in recent years. A few studies have been published to address this issue in online social networks, with the Arabic language receiving very little attention.

Aldjanabi et al. [6] used a multitask learning (MTL) model built on top of a pre-trained Arabic language model (BERT) to identify offensive and hate speech on social media. They trained the MTL model in three different Arabic offensive and hate speech datasets: (1) OSACT [7], containing two datasets with 10,000 samples each; (2) L-HSAB [8], containing 6024 samples; and (3) T-HSAB [9], containing 5846 samples. Their experiments showed that the MLT model outperforms the single-task model with 95.20% accuracy and a 92.34% F1 score.

Alsafari and Sadaoui [10] used an Arabic hate speech dataset containing 9338 tweets as a SEED to automatically label their dataset, which contained 5 M unlabeled tweets, using the semi-supervised self-training method. They used the deep learning algorithms of machine learning (i.e., support vector machine (SVM), CNN, and BiLSTM) along with text vectorization techniques (i.e., n-gram, word2vec skip-ram, AraBERT, and DistilBERT) to conduct experiments. Their results showed the superiority of the BiLSTM algorithm and World2Vec Skip-Gram text vectorization in terms of precision (88.89%), recall (89.83%), and F1 score (89.35%).

Muaad et al. [11] proposed a model for misogyny detection in Arabic tweets using two datasets [12]. They conducted seven machine and deep learning techniques. The first dataset contained 7866 tweets for binary classification, while the second one contained 6550 tweets for multiclassification. The results showed that AraBERT outperforms other algorithms with 90% and 89% accuracies for binary and multiclass classifications, respectively.

Meanwhile, Alshalan and Al-Khalifa [13] built an Arabic hate dataset containing 9316 tweets and used the four models of CNN, GRU, CNN + GRU, and BERT to evaluate detection performance. Their results showed that the CNN model performs best, with a 79% F1 score and an 89% area under the receiver operating characteristic curve.

Albayari and Abdallah [14] collected an Arabic cyberbullying dataset from Instagram with four different dialects: Modern Standard Arabic (MSA), Egyptian, Levantine, and Gulf. This dataset contained 46,898 comments, which were classified as 12,256 bullying comments, 5937 toxic comments, 17,376 positive comments, and 11,329 neutral comments. They utilized four machine learning algorithms (i.e., multinomial naïve Bayes, logistic regression, random forest, and SVMs) to evaluate the detection process performance. The experimental results showed that SVMs outperform other algorithms with 69% accuracy.

Althobaiti [15] proposed an automatic method for detecting offensive language and precise hate speech in Arabic tweets. They used a dataset [16] with 12,698 tweets classified into 8235 clean and 4463 offensive tweets. They also investigated the use of sentiment analysis and emoji descriptions as appending features along with the textual content of the tweets. Two machine learning algorithms (i.e., SVM and logistic regression) and a deep learning algorithm (i.e., BERT) were employed to evaluate the classification performance. The results showed that BERT outperforms other algorithms with an 84.3% F1 score.

Table 1 presents a list of datasets pertaining to cyberbullying along with pertinent information about each dataset. The first column contains the study or source of the dataset. The second column specifies the platform from which the data was collected, such as Formspring, Twitter, Instagram, MySpace, etc. The third column indicates the language of the data, which is predominantly English but also includes Dutch and Japanese. The fourth column displays the size of each dataset, which ranges from 1340 to 1,570,000 instances. Some datasets are balanced, indicating that the proportion of positive and negative instances is roughly equal, while others are imbalanced, with a higher proportion of one class compared to the other. The fifth column provides details about the balancing of each dataset, expressed as a decimal value between 0 and 1. For instance, a value of 0.142 in the balancing column signifies that 14.2% of instances in the dataset belong to the positive class, while the remaining 85.8% belong to the negative class. Table 1 offers valuable information for researchers interested in studying cyberbullying and developing machine learning models to detect and prevent it.

**Table 1.** Cyberbullying datasets.

| Study | Platform | Language | Size | Balancing |
|---|---|---|---|---|
| [17] | Formspring | English | 3915 | 0.142 |
| [18] | YouTube Formspring | English | - | - |
| [19] | Twitter, MySpace | English | 1,570,000 | - |
| [20] | YouTube | English | 4626 | 0.097 |
| [21] | Twitter | English | 4865 | 0.019 |
| [22] | Kaggle | English | 2647 | 0.272 |
| [23] | Twitter | English | 1340 | 0.152 |
| [24] | Ask FM | Dutch | 85,485 | 0.067 |
| [25] | Schoolboard Bulletins (BBS) | Japanese | 2222 | 0.128 |
| [26] | Twitter | English | 4865 | 0.186 |
| [27] | Twitter | English | 10,007 | 0.06 |
| [28] | Twitter | English | 1762 | 0.388 |
| [29] | Train-Formspring and MySpace Test-Twitter | English | 3279 | 0.12 |
| [30] | Instagram | English | 1954 | 0.29 |
| [31] | Formspring | English | 13,000 | 0.066 |
| [32] | Formspring | English | 13,160 | 0.194 |

Table 2 presents a specific focus on prior research conducted in the realm of Arabic cyberbullying detection. The first column comprises the study or source of the dataset. The second column delineates the dataset utilized in the study, encompassing Twitter in Arabic and English, as well as Aljazeera.net. The third column designates the feature representation employed in the study, including SentiStrength Feature Vector, word embeddings, TF-IDF, and n-gram. The fourth column enumerates the classifiers utilized in each study, encompassing naïve Bayes, SVM, KNN, random forests, logistic regression, convolutional neural networks (CNN), and recurrent neural networks (RNN). The fifth column presents the performance metrics of each classifier, such as accuracy (Acc), precision (P), recall (R), and F1 score (F), which are commonly utilized in machine learning to assess the quality of a classifier's predictions. Table 2 offers a comprehensive overview of the various approaches adopted in Arabic cyberbullying detection studies along with their corresponding performance metrics, thereby serving as a valuable resource for researchers engaged in this field.

**Table 2.** Previous work in Arabic cyberbullying detection.

| Study | Dataset | | | Feature Representation | Classifier | Performance | | | |
|---|---|---|---|---|---|---|---|---|---|
| | Platform | Size | Classes | | | Acc | P | R | F |
| [33] | Twitter | Arabic = 35,273 English = 91,431 | Yes/No | Tweet to SentiStrength Feature Vector | Naïve Bayes SVM | | 93.4 | 94.1 | 92.7 |
| [34] | Twitter | Large = 34,890 Small = 4913 | Yes/No | Word embedding | FFNN | 94.5 | | | |
| [35] | Twitter | 34,890 | Bully Non-bully | | Bagging, boosting (KNN, SVM, NB) | | 93.3 | 93.5 | 92.0 |

**Table 2.** *Cont.*

| Study | Dataset | | | Feature Representation | Classifier | Performance | | | |
|-------|---------|---|---|---|---|---|---|---|---|
| | **Platform** | **Size** | **Classes** | | | **Acc** | **P** | **R** | **F** |
| [36] | Twitter | Real-Time Classification | | TF-IDF | | | | | |
| [37] | YouTube and Twitter | 25,000 | | TF-IDF | Naïve Bayes (NB) | 95.9 | 92.9 | 92.5 | 92.7 |
| [38] | YouTube Twitter | training (100,327) testing (2020) | | TF-IDF | PMI, Chi-square Entropy | 81.0, 62.1, 39.1 | | | |
| [39] | Aljazeera.net. (test) Twitter | 32K | CB, NCB | Word embedding TF-IDF, n-gram, Bow | CNN, RNN | | | | 84.0 |
| [40] | Facebook and Twitter. | 6138 | Positive/ Negative | TF-IDF | KNN, SVM, NB, random forests, and J48 | | 94.5 | 94.4 | 94.4 |
| [41] | Twitter | | Bullying/No bullying | Sentiment analysis, emojis and user history | | 85.0 | | | |
| [42] | Twitter | 151,000 | | Sentiment analysis | Ridge regression (RR) and logistic regression (LR) | | | | |

## 3. Background

In this study, we utilized various deep learning methods, including LSTM, BiLSTM, CNN, GRU, BiGRU, AraBERT, and mBERT models, along with word embeddings. These techniques were implemented using the Keras neural network library for the purpose of classifying Arabic toxic tweets. The aim was to validate our proposed dataset through this comprehensive approach.

### 3.1. LSTM

LSTM is a recurrent neural network (RNN) type that can learn long-term dependencies. LSTM has a chain-like structure similar to an RNN, but its base module is structurally different from other RNNs. It has the advantage of having a cell state that stores and converts the input cell memory to the output cell state. The input, output, forget, and update gates are the main components of the LSTM cell architecture. The forget gate determines what information should be forgotten from previous memory units. The input gate determines what information should be accepted into the neuron. The output gate generates a new long-term memory. Lastly, the update gate brings the cell up to date. These four components work together in a certain way, accepting short- and long-term memories and input sequences at a given timestamp and forming new long- and short-term memories and output sequences at that same timestamp [43]. The BiLSTM is an extension of the traditional LSTM that processes the sequence in both directions [44].

### 3.2. CNNs

A CNN is an advanced, high-potential, classical artificial neural network model that can handle higher-complexity data and tough data compilation and preprocessing. It is one of the most efficient and powerful models for processing image and nonimage data, which is based on the neuron configurations in the visual cortex of an animal's brain. CNNs are primarily used in computer vision. However, they are also characterized by classification issues in natural language processing. A CNN is especially suitable for retrieving new

attributes from short fixed-length blocks throughout a dataset, regardless of the feature locations. Convolutional, pooling, and fully linked layers are included. Some filters (kernels) slide across the preprocessed signals in the convolutional layer, and the feature map is obtained after the convolution procedure. The second layer uses a pooling technique to minimize the dimension of the standard layer output, thereby preventing overfitting and lowering the computing intensity. The final layer is concerned with activation functions, which make the output nonlinear [43].

### 3.3. GRU

The GRU addresses the drawbacks of the RNNs (e.g., vanishing gradient problem) by utilizing the update and reset gate methods. The update and reset gates are vectors that specify what data should be supplied to the output unit. The most interesting feature of the GRU is that it can be properly taught to preserve information for a long time without losing track of timestamps. BiGRU is a sequence processing model with two GRUs. One goes forward with information, while the other goes backward with it. This bidirectional RNN only has input and forget gates [45].

### 3.4. AraBERT

AraBERT is based on BERT. BERT is a language representation model released by Google [21], and it is widely accepted as the basis for the latest results of various NLP tasks in multiple languages. It is currently one of the most used languages in modeling architectures. Its generalization features enable it to be tailored to various downstream tasks dependent on user needs (i.e., whether it is named-entity recognition, relation extraction, question answering, sentiment analysis, or sequence classification) [46].

The mBERT model is trained in many languages, including Arabic, and is a universal language model. The top 104 languages, including Wikipedia's dataset, were used to train the mBERT model. Multilingual models such as mBERT are usually outperformed by monolingual models pre-trained with a larger vocabulary and bigger language-specific datasets [47].

## 4. Methodology

The proposed approach encompasses three main phases: dataset creation, preprocessing, and classification. Figure 1 provides a summary of the steps involved in the methodology and the pseudocode is listed in Algorithm 1.

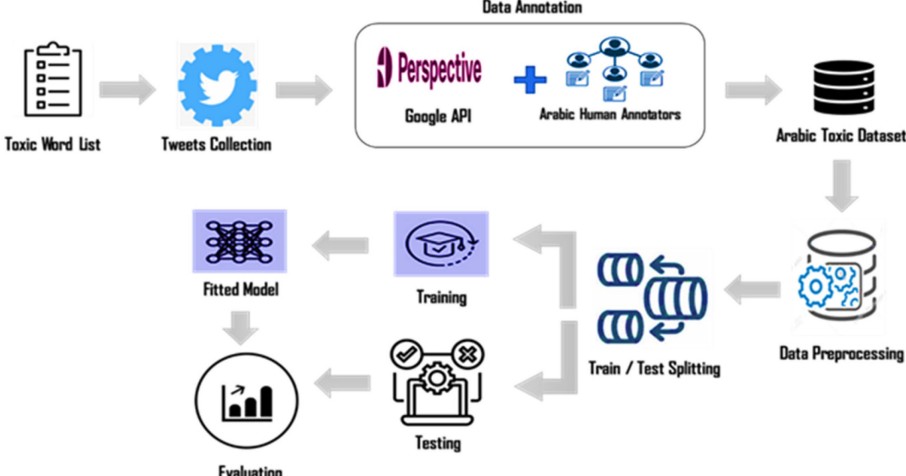

**Figure 1.** Summary of methodology steps: Dataset creation, preprocessing, and classification.

The following pseudocode shows the methodology's steps:

---

**Algorithm 1:** Methodology Steps: Dataset Creation, Preprocessing, and Classification.

---

**Input**: Arabic toxic word list
**Output**: Tweet label (toxic, non-toxic)

1. Load the Arabic toxic word list.
2. Initialize an empty list: collected_tweets.
3. Use the Twitter API to collect Arabic tweets containing words from the toxic word list:
   - For each word in the toxic word list:
     ○ Search the Twitter API for tweets containing the current word.
     ○ Add the collected tweets to the collected_tweets list.
4. Initialize two empty lists: manually_annotated_toxic and manually_annotated_non-toxic.
5. For each tweet in the collected_tweets list:
   - Use Google's Perspective API to determine the toxicity score of the tweet.
   - If the toxicity score is 0.35 or greater, add the tweet to the manually_annotated_toxic list.
6. For each tweet in the manually_annotated_toxic list:
   - Have three Arabic speakers manually annotate the tweet as toxic or non-toxic.
   - Add the annotated tweet to the manually_annotated_non-toxic list.
7. Combine the manually_annotated_toxic and manually_annotated_non-toxic lists into a single list called annotated_tweets.
8. For each tweet in the annotated_tweets list:
   - Clean the tweet by removing unnecessary characters and formatting.
   - Remove stop words from the cleaned tweet.
   - Normalize the cleaned tweet (e.g., convert to lowercase, remove diacritics).
9. Split the annotated_tweets into features (preprocessed tweets) and labels (toxic/non-toxic) (using a random state in Python's scikit-learn library to split the data into training and testing sets is a good practice to ensure that the data is truly random and unbiased)
10. Train a classification algorithm using the preprocessed tweets as features and the corresponding labels as targets.
11. Calculate classification performance metrics using the trained model:
12. Display the calculated performance metrics.

**Output**: Displayed classification performance metrics (accuracy, precision, recall, F1 score).

---

### 4.1. Dataset Creation Phase

This phase included two main steps: dataset building and annotation. Insulting texts frequently appearing in social media messages are excellent signs of toxicity. Different texts and behaviors (e.g., insulting, abusing, provoking, and aggressive behavior) can lead to toxicity. Omar et al. [48] provided a list of ten key definitions for Arabic words used in hate speech. We updated this list and used a mix of Egyptian and Gulf keywords covering ten categories. Twitter API [49] was used to collect the dataset. A search was made for each word in the keyword list, and all returned tweets were saved. Some non-toxic keywords and hostages were also used to collect normal tweets. After all the searches, duplicate tweets and those containing only one word were removed. The number of final collected tweets was 40,000 (https://github.com/AhmedCS2015/Arabic-Toxic-Dataset (accessed on 29 August 2023)). Table 3 presents samples of the keywords used and tweets and the corresponding toxic category of each word.

The annotation phase was divided into two steps. We employed Google's Perspective API [50], developed by Google's Project Jigsaw and "Counter Abuse Technology" teams, as a first step toward toxic tweet identification. Google's Perspective API was highly effective in multiple studies [51–53]. It provided a set of machine learning models for identifying how toxic/hateful a sentence is and supports a set of languages, including Arabic. The API returned a probability score between 0 and 1. The higher values indicated a greater likelihood of the toxicity label being applied to the text. The toxicity scores were based on a probability score of 0 to 1; hence, we selected toxic tweets with scores of 0.35 or greater in the first steps of our annotation process. This step filtered the collected

tweets to 17,000 toxic tweets. Table 4 shows the tweet count percentage based on Google's Perspective API.

**Table 3.** Samples of toxic tweets.

| Ar-Keyword | En-Keyword | Translation | Sample Tweet |
| --- | --- | --- | --- |
| ساقط | Fallen | You are impolite, uneducated, paid, and a fallen Baathist | انت غير مؤدب وغير مثقف ومأجور وبعثي ساقط |
| قبيح | Ugly | God does not give you anything nice, an ugly face, a bad tongue, malicious eyes. | ربنا مش مديك اى حاجة حلوه وجه قبيح لسان عبيط عيون خبيثة |
| زبال | Trashy | Valverde is the least-losing coach in Barcelona's history, scavenger | فالفيردي المدرب الأقل خسارة في تاريخ برشلونة يا زبال |
| بزر | Insignificant | Make a mistake about your Muslim brother because of cardboard, kid. | تغلط علا اخوك المسلم عشان كرتون يا بزر |
| سافل | Damn | Damn grant that's bad | منحط سافل هذا ردي |
| خنزير | Pig | What is the saying of a donkey, a pig, a genus that is human beings? | علماني إيه قول حمار خنزير خرتيت أي جنس غير إنه يكون بني آدم |
| مريض | Sick | My brother, by God, you are sick and do not understand | ياخوي والله انك مريض وماتفهم |

**Table 4.** Sample tweets and their toxicity level percentages.

| Toxicity Percentage | Tweet Count |
| --- | --- |
| ≥0.5 | 3470 |
| ≥0.35 and <0.5 | 9400 |
| ≥0.4 | 6600 |
| ≥0.35 and <0.4 | 6200 |

We have developed an online platform (as shown in Figure 2) that provides flexible text annotation capabilities with full-screen mode and customizable buttons. This platform allows experts to access it at any time and from any device, providing them with convenience and flexibility in completing their assigned tasks.

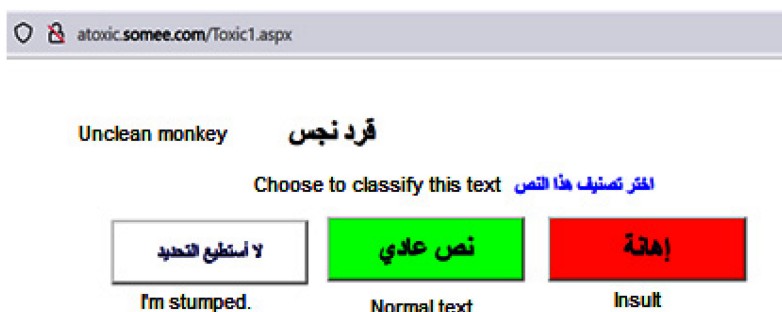

**Figure 2.** Online platforms for manual annotation.

Key features of our online platform include:

1. Full-screen mode: The text is displayed in a full-screen view, maximizing the visibility and readability of the content.
2. Customizable buttons: Three buttons are provided, allowing experts to easily select the type of annotation they want to apply to the text.

By offering these features, our platform aims to enhance the annotation process, making it more efficient and user-friendly for experts.

As we discuss the annotation process, it's important to consider the potential mental health implications for the experts exposed to a substantial number of toxic tweets. It's noteworthy that our approach aims to mitigate any undue stress by allowing experts the flexibility to annotate at their own pace. They are not required to complete the annotation in one sitting or within a specific timeframe. Our online platform permits experts to access and annotate tweets at their convenience, fostering a more manageable and considerate workflow. This flexibility allows experts to engage with the annotation process in a manner that aligns with their well-being, minimizing potential negative impacts on mental health.

Before the annotation process, there are tweets that contain keywords but do not represent toxicity, such as:

- I love raising dogs.
- The teacher is sick today.

The garbage collector comes on time every day.

By using the Google API annotation process, these tweets receive low scores, which makes the manual categorization process easier for us. The Google API assists us in filtering out toxic tweets and pinpointing those with significant toxic scores, thereby facilitating the manual annotation process.

### 4.2. Preprocessing Phase

We first performed several preprocessing steps on the dataset before feeding the tweets as inputs to the classification models. Preprocessing is essential when working with noisy and informal data such as those from Twitter. Given the inherited ambiguity of the Arabic language and the wide range of vernacular Arabic used on Twitter, this assumes even greater significance for the Arabic language. The preprocessing steps include the following:

- Cleaning: Punctuation, additional whitespace, diacritics, and non-Arabic characters are eliminated in this step.
- Stop word elimination: Many terms in the text-preprocessing task have no essential meaning but are used frequently in a document. They do not help improve the performance because they do not provide much information for the classification task. Stop words should be eliminated before the feature selection process.
- Normalization: Many normalization methods are used, including stemming, to make all words acquire the same form. We can perform normalization using various techniques (e.g., regular expressions). The normalization steps are as follows:
- Different forms of "ا" ("أ," "إ," and "آ") are replaced with "ا."
- "ئ" and "ى" at the end of the word are replaced by "ي."
- "ه" at the end of the word is replaced by "ة."
- Repeated letters are replaced with a single letter (e.g., "جووووول" converted to "جول" it means (Goal)).

### 4.3. Classification Phase Using the AraBERT Model

As a trained language model, AraBERT formed a black box with prior knowledge of a natural language that may be used and fine-tuned to handle various NLP challenges. The pretraining method employed inexpensive unlabeled data to learn the initial neural network model parameters.

In AraBERT, each tweet was first tokenized to N tokens, and a special classification token [CLS] was added at its beginning. The model then generated an embedding representation (Ei) for each token, i. The token-level representations were calculated using information from the entire sentence by the sum of the following three value vectors: (1) the vector embeddings corresponding to the token; (2) the segment embedding vector (i.e., the tweet to which it belongs); and (3) the token position vector (i.e., token position inside the tweet) (Figure 3).

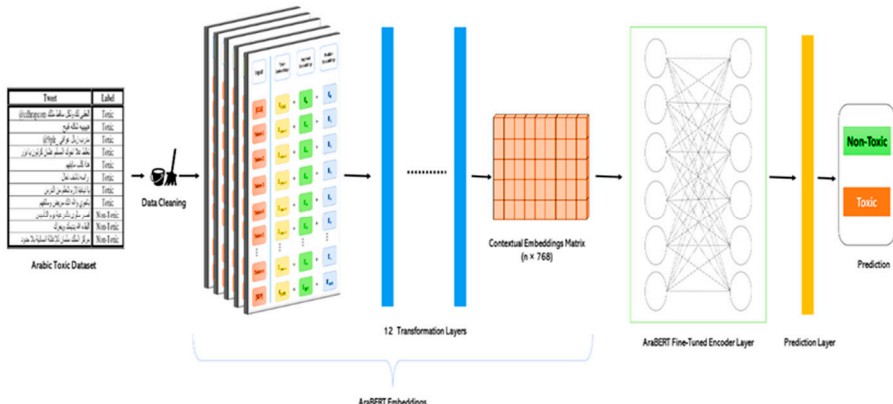

**Figure 3.** Toxic tweet classification pipeline.

The resulting token representation was fed through twelve transformer layers comprising encoders and decoders. An encoder was built using multi-head attention and feed-forward neural network science. Transformers heavily rely on attention, and their self-awareness gains a contextual understanding of words in text based on adjacent words in a sentence. The AraBERT embedding was 768 length, which is a common value for many NLP tasks; hence, each word in the input item was converted into a vector of $1 \times 768$ dimensions. This method was performed for each n-word in a tweet, resulting in a contextual matrix of $n \times 768$. AraBERT provided contextualized sentence-level representations that helped classifiers better understand sentence semantics [46].

In actuality, the input tweet contained fewer than n words since AraBERT appended tokens to denote the start of each sentence and the end of each sentence. The tweet length was similarly limited to L words, with L being any number. Tweets that were fewer than L words in length were filled by adding 0 vectors at the end. The input was processed, tokenized, padded, and converted to a PyTorch tensor. AraBERT embedding was then used to convert the text to embed [46].

### 4.4. Model Training and Fine-Tuning

The AraBERT model was pre-trained on general corpora, and we dealt with Twitter content in our toxic detection task. As a result, one critical step was to assess the contextual information collected from AraBERT's pre-trained layers and fine-tune the layers using our annotated datasets.

The AraBERT model was trained in two phases. The first phase involved pretraining the AraBERT language model, while the second phase included fine-tuning the outmost classification layer. The AraBERT model was trained using the MSA data. AraBERT has another version (AraBERTv0.1), but we used AraBERTv0.2-Twitter-base because it was designed for Arabic dialects and tweets and trained by continuing the pretraining through the masked language modeling task on ~60 M Arabic tweets filtered from a collection of 100 M. This version also contained common words that were not in the first versions. The pretraining was done with a max sentence length of 64 for one epoch [46].

After using the AraBERT model architecture for classification, the toxic text features were integrated into the model. At this stage, the results were fine-tuned for the toxic tweet classification task using different characteristic indicators. The fully connected classification layer was trained in the same manner. The categorical cross-entropy was utilized as the loss function during training. The toxic tweet probability was calculated using the abovementioned model at the output stage. The sigmoid activation function was also used to obtain the identification result (i.e., toxic or non-toxic).

### 5. Results

We applied a fine-tuned AraBERT model to classify toxic Arabic tweets by implementing the following deep learning algorithms: CNN, LSTM, BiLSTM, GRU, and BiGRU with

AraBERT and mBERT embeddings. Evaluating various implemented deep learning and pre-trained models revealed that the AraBERT algorithm outperforms other models. Model training was conducted on 70% of randomly selected data points, tested on 15% of data, and validated on 15% of data. The model performances were evaluated based on the metrics of accuracy, recall, precision, and F1 score. All models were compared, and these comparisons were evaluated and visualized. Table 5 depicts the comparative analysis results of the dataset using deep learning models. This section presents the experiment results.

**Table 5.** Toxic tweet classification results using deep learning models.

| Model | Accuracy | Precision | Recall | F1 Score |
|---|---|---|---|---|
| LSTM | 0.9767 | 0.9623 | 0.9831 | 0.9763 |
| GRU | 0.9806 | 0.9796 | 0.9828 | 0.9803 |
| BiLSTM | 0.9810 | 0.9728 | 0.9851 | 0.9812 |
| BiGRU | 0.9860 | 0.9838 | 0.9880 | 0.9870 |
| RNN | 0.9872 | 0.9865 | 0.9890 | 0.9866 |

Table 5 and Figure 4 present the outcomes of the toxic tweet classification process, employing various deep learning models. The evaluation metrics include accuracy, precision, recall, and F1 score, all of which play a crucial role in gauging the performance of each model. The LSTM model demonstrates a commendable overall performance, achieving an accuracy of 0.9767. It maintains a solid balance between precision (0.9623) and recall (0.9831), leading to an impressive F1 score of 0.9763.

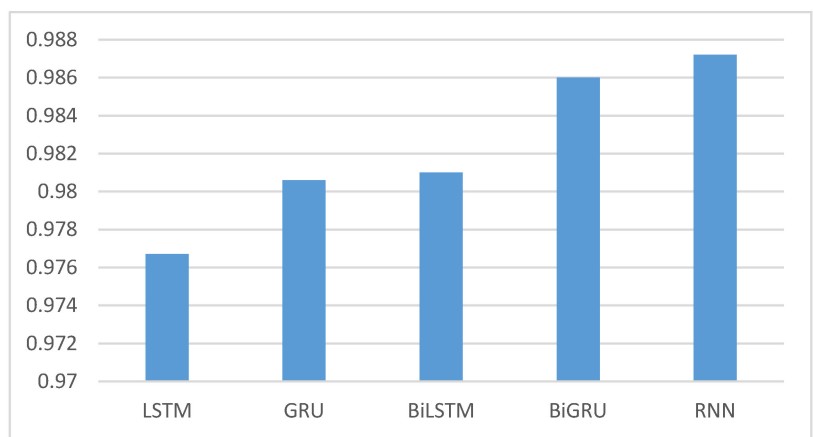

**Figure 4.** Results of toxic tweet classification utilizing deep learning models.

Similarly, the GRU model showcases notable results, with an accuracy of 0.9806. This model excels in both precision (0.9796) and recall (0.9828), contributing to an elevated F1 score of 0.9803. The BiLSTM model continues to exhibit strong classification capabilities, achieving an accuracy of 0.9810. With a precision of 0.9728 and a high recall of 0.9851, it attains an F1 score of 0.9812. The BiGRU model stands out as one of the top performers, securing an accuracy of 0.9860. It achieves an exceptional balance between precision (0.9838) and recall (0.9880), resulting in a remarkable F1 score of 0.9870.

Lastly, the RNN model showcases the highest accuracy among the models, reaching 0.9872. With precision and recall values of 0.9865 and 0.9890, respectively, it sustains a robust F1 score of 0.9866.

The results underline the effectiveness of deep learning models in classifying toxic tweets, with each model displaying strengths in different aspects of classification performance. The BiGRU and RNN models, in particular, emerge as the most potent choices with the highest accuracy and balanced precision–recall–F1 scores.

Table 6 and Figure 5 show the results of using AraBERT and mBERT as word embeddings with deep learning models.

**Table 6.** Toxic tweet classification results using word embeddings with deep learning models.

| Word Embedding | Model | Accuracy | Precision | Recall | F1 Score |
|---|---|---|---|---|---|
| AraBERT | LSTM | 0.9934 | 0.9923 | 0.9944 | 0.9930 |
| | GRU | 0.9905 | 0.9895 | 0.9908 | 0.9900 |
| | BiLSTM | 0.9917 | 0.9905 | 0.9923 | 0.9911 |
| | BiGRU | 0.9888 | 0.9803 | 0.9901 | 0.9815 |
| | RNN | 0.9868 | 0.9808 | 0.9898 | 0.9838 |
| mBERT | LSTM | 0.9242 | 0.9222 | 0.9262 | 0.9232 |
| | GRU | 0.9264 | 0.9234 | 0.9272 | 0.9254 |
| | BiLSTM | 0.9252 | 0.9232 | 0.9284 | 0.9241 |
| | BiGRU | 0.9244 | 0.9235 | 0.9252 | 0.9241 |
| | RNN | 0.8723 | 0.8712 | 0.8742 | 0.8720 |

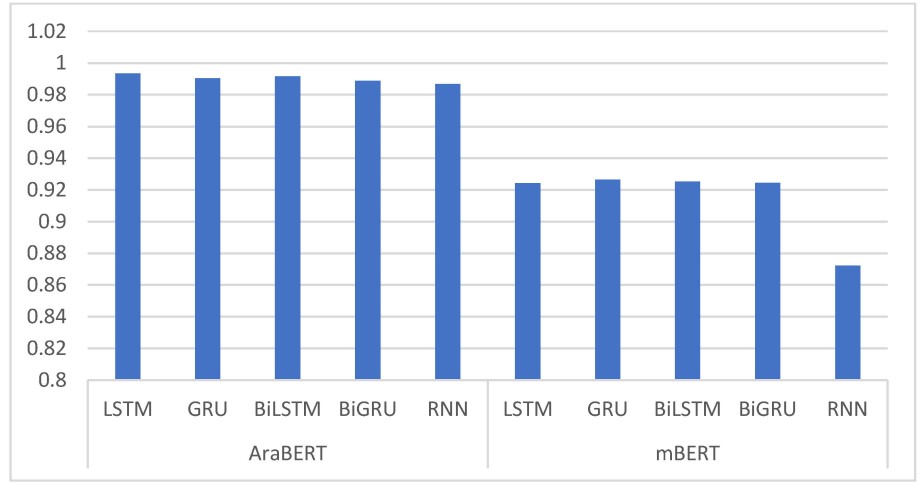

**Figure 5.** Classification results of toxic tweets using deep learning models with word embeddings.

Table 6 and Figure 5 provide an overview of the toxic tweet classification outcomes achieved by utilizing word embeddings in conjunction with deep learning models. The LSTM model, implemented with the AraBERT word embedding, showcases exceptional classification prowess. With an F1 score of 0.9930, a recall of 0.9944, a precision of 0.9923, and an accuracy of 0.9934, it demonstrates robust classification capabilities. Similarly, the GRU model, also utilizing AraBERT word embedding, achieves a high F1 score of 0.9900, along with a recall of 0.9908, a precision of 0.9895, and an accuracy of 0.9905. The BiLSTM model with the same AraBERT word embedding maintains a strong classification performance, securing an F1 score of 0.9911, a recall of 0.9923, a precision of 0.9905, and an accuracy of 0.9917. The BiGRU model, implemented with AraBERT word embedding, achieves an F1 score of 0.9815, a recall of 0.9901, a precision of 0.9803, and an accuracy of 0.9888.

Incorporating mBERT word embedding, the LSTM model attains an F1 score of 0.9232, a recall of 0.9262, a precision of 0.9222, and an accuracy of 0.9242. Similarly, the GRU model with mBERT word embedding demonstrates an F1 score of 0.9254, a recall of 0.9272, a precision of 0.9234, and an accuracy of 0.9264. The BiLSTM and BiGRU models with mBERT word embedding yield similar results, achieving F1 scores of 0.9241 and 0.9241, respectively, along with recall, precision, and accuracy scores in the same range. The RNN model, implemented with both AraBERT and mBERT word embeddings, achieves an F1 score of 0.8720, indicating its slightly lower classification performance compared to the other models.

The utilization of word embeddings in tandem with deep learning models showcases varying degrees of classification proficiency. The AraBERT-based models generally perform

exceptionally well across all metrics, highlighting the importance of word embeddings in enhancing toxic tweet classification.

Table 7 and Figure 6 present the fine-tuned pre-trained models.

**Table 7.** Toxic tweet classification results using fine-tuned pre-trained models.

| Model | Accuracy | Precision | Recall | F1 Score |
| --- | --- | --- | --- | --- |
| AraBERT | 0.996037 | 0.99604 | 0.996035 | 0.996037 |
| mBERT | 0.98689 | 0.9869 | 0.986895 | 0.98689 |

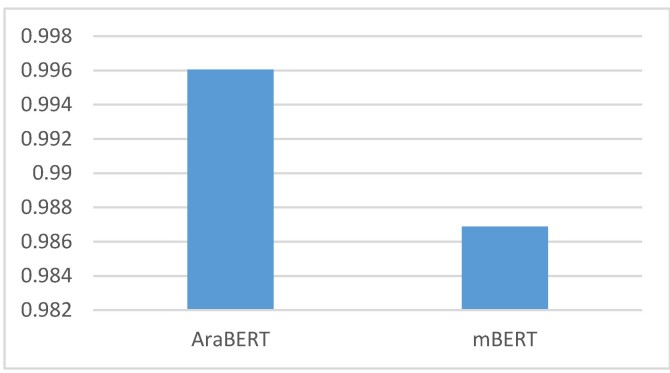

**Figure 6.** Fine-tuned pre-trained models' accuracy.

Notably, the AraBERT model outperforms in this classification task, showcasing exceptional performance across all evaluation metrics. With an impressive accuracy of 0.996037, the AraBERT model not only achieves a high level of correct predictions but also demonstrates its capability in minimizing misclassifications. This superior performance is further highlighted by its precision of 0.99604, recall of 0.996035, and F1 score of 0.996037, all of which consistently indicate the model's proficiency in accurately identifying toxic tweets.

The superior performance of the AraBERT model emphasizes its effectiveness in capturing nuanced linguistic patterns associated with toxic content. Its ability to achieve such high precision, recall, and F1 scores underscores its potential to both detect and distinguish toxic tweets with remarkable accuracy. Figure 7 illustrates the confusion matrix for the AraBERT model.

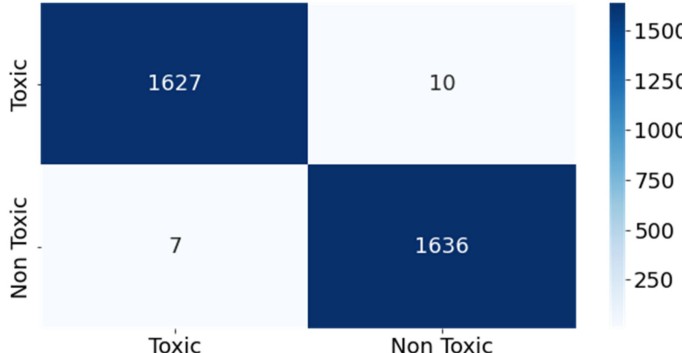

**Figure 7.** Confusion matrix of AraBERT.

We intentionally balanced the dataset to avoid imbalance issues (which does not represent the natural distribution of toxic tweets compared to non-toxic ones). However, for the optimal training of a model, it is beneficial to have a balanced dataset. We performed undersampling of non-toxic tweets to achieve this balance.

Based on the results reported in Tables 5–7, I would recommend the AraBERT model for toxic tweet classification. In Table 5, which evaluates different deep learning models without word embeddings, the BiGRU model has the highest F1 score of 0.9870.

However, in Table 6, where word embeddings are incorporated, all models using the AraBERT embedding substantially outperform the others. The LSTM model achieves the highest F1 score of 0.9930 when using AraBERT. Most significantly, as shown in Table 7, the AraBERT model alone achieves an exceptional F1 score of 0.996037 when fine-tuned, far surpassing the mBERT model. Considering that F1 score is the primary metric recommended for imbalanced datasets like this one, the AraBERT model stands out as the top performer. It demonstrates high precision and recall in addition to high F1 scores.

Therefore, based on its consistently superior results across evaluation, I would recommend the pre-trained AraBERT model for the task of toxic tweet classification. The substantial gains over other methods, especially in F1 scores, suggest it can most accurately identify toxic content while balancing precision and recall.

## 6. Discussion

Social media platforms have revolutionized communication and information sharing, enabling interactive exchanges among users on a massive scale. However, alongside the positive aspects, these platforms have also become breeding grounds for the dissemination of inappropriate and toxic content, including hate speech and insults. While substantial efforts have been dedicated to classifying toxic content in the English language, Arabic texts have not received the same level of attention. This study aims to bridge this gap by constructing a standardized Arabic dataset explicitly tailored for the classification of toxic tweets.

To ensure the reliability and quality of the dataset, we employed a two-step annotation process. First, we utilized Google's Perspective API, a powerful tool for automated toxicity detection, to annotate a substantial portion of the dataset. However, recognizing the nuances and unique characteristics of Arabic language, we sought the expertise of three native Arabic speakers and linguists. Their manual annotations refined the dataset, ensuring a more accurate representation of toxic content in Arabic tweets.

To evaluate the performance of various models in classifying toxic tweets, we conducted a series of experiments using seven different models: long short-term memory (LSTM), bidirectional LSTM, a convolutional neural network (CNN), a gated recurrent unit (GRU), bidirectional GRU, multilingual bidirectional encoder representations from transformers (mBERT), and AraBERT. Additionally, we leveraged word embedding techniques to enhance the models' understanding of semantic relationships between words.

Our experimental findings revealed that the fine-tuned AraBERT model consistently outperformed other models in terms of classification accuracy. Notably, the AraBERT model achieved an impressive accuracy of 0.9960, surpassing the performance of the other models. This high accuracy demonstrates the effectiveness of AraBERT in accurately identifying toxic content in Arabic tweets. Importantly, our results also surpass the accuracies reported in recent literature for similar approaches, highlighting the significance of our study's contribution.

This study represents a significant advancement in the field of Arabic toxic tweet classification. By addressing the need for a standardized Arabic dataset and leveraging state-of-the-art deep learning models, we have shed light on the importance of combating toxicity in social media platforms while considering diverse languages and cultures. The findings of this study pave the way for future research and practical applications in developing robust and efficient systems for monitoring and mitigating toxic content in Arabic social media.

However, it is important to acknowledge certain limitations in our study. The dataset, although carefully constructed and annotated, may not capture the entire spectrum of toxic content due to the ever-evolving nature of language and emerging trends in online toxicity. Additionally, the performance of the models may vary when applied to different domains or contexts. Future research should focus on expanding the dataset and exploring novel techniques to improve the models' generalizability and adaptability.

## 7. Limitations

This study has several limitations that should be acknowledged. Firstly, the focus of the study is specifically on toxic tweet classification in the Arabic language, neglecting other languages and types of toxic content such as images or videos. Therefore, the findings may not be applicable to different languages or content formats. Secondly, the automatic annotation of the dataset using Google's Perspective API introduces potential errors and limitations, as automated methods may not capture the full complexity and context of toxic content accurately. Human biases or limitations in the API's understanding of Arabic nuances could impact the quality of the dataset annotations. Thirdly, although the dataset is annotated by three native Arabic speakers and linguists, variations in their expertise and subjective judgments may influence the reliability of the dataset.

Despite these limitations, the study demonstrates promising results, with the fine-tuned AraBERT model outperforming other models, highlighting its advancement in Arabic toxic tweet classification and emphasizing the importance of addressing toxicity while considering diverse languages and cultures on social media platforms.

## 8. Conclusions

This study presents a comprehensive approach to tackling toxic content on social media platforms, specifically focusing on Arabic toxic tweet classification. We have developed a dataset consisting of 31,836 tweets, annotated through a combination of automated and manual processes involving Google's Perspective API and the expertise of native Arabic speakers and linguists. Our exploration encompassed a range of deep learning models, word embeddings, and fine-tuned pre-trained models, evaluating their performance across various metrics. The results highlighted the exceptional capabilities of the AraBERT model, which consistently outperformed other models with remarkable accuracy, precision, recall, and F1 scores. This study emphasizes the critical role of model selection and contextual considerations in achieving accurate toxic tweet classification. Moving forward, there are opportunities to expand the dataset, incorporate multiple labels for improved detection, explore different embedding techniques, and address dialectics and diverse topics. By leveraging advanced techniques and methodologies, we can continue to enhance the accuracy and effectiveness of toxic content classification in the ever-evolving landscape of text analysis.

We believe that several methods can be employed to extend and improve this study in the future. Our dataset can be extended to capture more dialectics, patterns, and topics. It can also be annotated with multiple labels to improve the detection results beyond binary classification. Moreover, different embedding techniques could be investigated, including Word2Vec ELMo, the Universal Sentence Encoder, and FastText.

**Author Contributions:** This work was carried out in collaboration among all authors. All Authors designed the study, performed the statistical analysis, and wrote the protocol. Authors A.O. and T.A.E.-H. managed the analyses of the study, managed the literature searches, and wrote the first draft of the manuscript. All authors have read and agreed to the published version of the manuscript.

**Funding:** This work was supported by the Deanship of Scientific Research, Vice Presidency for Graduate Studies and Scientific Research, King Faisal University, Saudi Arabia [Project No.: GRANT4,055].

**Institutional Review Board Statement:** Not applicable.

**Informed Consent Statement:** This article does not contain any studies with human participants or animals performed by any of the authors.

**Data Availability Statement:** The dataset and code used in this study is public and all test data are available at this portal (https://github.com/AhmedCS2015/Arabic-Toxic-Dataset (accessed on 29 August 2023)).

**Conflicts of Interest:** The authors declare no conflict of interest.

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
