# Peer review of "Arabic Toxic Tweet Classification: Leveraging the AraBERT Model"

_2504-2289, doi:10.3390/bdcc7040170_

Round 1

Reviewer 1 Report

The paper, titled "Arabic Toxic Tweet Classification: Leveraging the AraBERT Model," explores the utilization of various deep learning models, particularly the AraBERT model, for the classification of toxic tweets. I believe this work makes a noteworthy contribution to the literature, primarily by curating a publicly accessible dataset consisting of 31,836 tweets that have been annotated as either toxic or non-toxic. However, I must emphasize that this paper requires substantial improvements before it can be considered for acceptance.

1.     Dataset Assembly: The most significant contribution of this paper lies in the assembly of a dataset for toxic Arabic tweets. I strongly recommend that the author provide more comprehensive details about the dataset creation process. For instance: a. Clarify the methodology employed by the three experts during the annotation of the dataset. b. Consider addressing the potential mental health implications for experts exposed to a large number of toxic tweets during the annotation process. c. Provide insights into the distribution of toxic scores from the Google API before and after annotation.

2.     Results and Performance: In the section discussing the performance results, it is not apparent that there is a substantial difference between various models. Given the likelihood of an imbalanced dataset (which I presume to be the case with the original tweets), I recommend that the paper place greater emphasis on F1 scores rather than accuracy scores. Based on the F1 scores, even the relatively lightweight RNN model does not exhibit a significant difference from the more complex models. This aspect may have limited practical utility in industrial settings in terms of both model training and maintenance costs.

3.     Dataset Sampling: I have concerns regarding the dataset, particularly regarding oversampling or undersampling. Figure 7 suggests a very balanced dataset, where the number of toxic tweets equals that of non-toxic tweets. This balance appears unusual, and while oversampling or undersampling can be performed during training, the testing set should ideally consist of the original data to ensure a more accurate evaluation.

Overall, the paper presents an important contribution in terms of the dataset it assembles for Arabic toxic tweets. However, addressing the aforementioned concerns and providing additional details regarding the dataset creation process, result interpretation, and dataset handling would significantly enhance the paper's quality and potential for acceptance.

Author Response

Big Data and Cognitive Computing

Special Issue "Advances in Natural Language Processing and Text Mining"

Manuscript Details

  • Manuscript ID: BDCC-2611406
  • Manuscript Title: “Arabic Toxic Tweet Classification: Leveraging the AraBERT Model".

Dear Editors,

We are providing a revised version of our paper following the comments and suggestions from the referees and also answering their questions. We have conducted a thorough revision of the paper following the received reviews. We want to thank the referees very much for their constructive criticisms which helped us to improve our paper. We also thank the editors for giving us the opportunity to revise our paper. Overall, we have included more detailed explanations and discussions. We also revised very carefully the presentation and organization of our paper to substantially improve clarity. Of course, we have also fixed the typos and grammar issues with the help of native English speakers. We have included a Word file for our paper with highlighted text for easy identification of the changes made. This will hopefully help to clarify that the paper contains sufficient new material compared to the related work research, to merit publication in this post-conference special issue. Below we explain in more detail how we addressed the referees’ comments.

We gratefully thank the editor and all reviewers for their time spent making their constructive remarks and useful suggestions, which have significantly raised the quality of the manuscript and enabled us to improve the manuscript. Each suggested revision and comment, brought by the reviewers was accurately incorporated and considered.

Sincerely,

The Authors

Reviewer comments and responses:

Reviewer #1

Reviewer Feedback

Response

1.    Dataset Assembly: The most significant contribution of this paper lies in the assembly of a dataset for toxic Arabic tweets. I strongly recommend that the author provide more comprehensive details about the dataset creation process. For instance: a. Clarify the methodology employed by the three experts during the annotation of the dataset.

Thank you for your feedback. We have developed an online platform (As shown in Figure 2) that provides flexible text annotation capabilities with full-screen mode and customizable buttons. This platform allows experts to access it at any time and from any device, providing them with convenience and flexibility in completing their assigned tasks.

Key features of our online platform include:

1.      Full-screen mode: The text is displayed in a full-screen view, maximizing the visibility and readability of the content.

2.      Customizable buttons: Three buttons are provided, allowing experts to easily select the type of annotation they want to apply to the text.

By offering these features, our platform aims to enhance the annotation process, making it more efficient and user-friendly for experts.

1.b. Consider addressing the potential mental health implications for experts exposed to a large number of toxic tweets during the annotation process

As we discuss the annotation process, it's important to consider potential mental health implications for the experts exposed to a substantial number of toxic tweets. It's noteworthy that our approach aims to mitigate any undue stress by allowing experts the flexibility to annotate at their own pace. They are not required to complete the annotation in one sitting or within a specific timeframe. Our online platform permits experts to access and annotate tweets at their convenience, fostering a more manageable and considerate workflow. This flexibility allows experts to engage with the annotation process in a manner that aligns with their well-being, minimizing potential negative impacts on mental health.

1.c. Provide insights into the distribution of toxic scores from the Google API before and after annotation.

Before the annotation process, there are tweets that contain keywords but do not represent toxicity, such as

·         I love raising dogs.

·         The teacher is sick today.

·         The garbage collector comes on time every day.

By using the Google API annotation process, these tweets receive low scores, which makes the manual categorization process easier for us. The Google API assists us in filtering out toxic tweets and pinpointing those with significant toxic scores, thereby facilitating the manual annotation process

2.    Results and Performance: In the section discussing the performance results, it is not apparent that there is a substantial difference between various models. Given the likelihood of an imbalanced dataset (which I presume to be the case with the original tweets), I recommend that the paper place greater emphasis on F1 scores rather than accuracy scores. Based on the F1 scores, even the relatively lightweight RNN model does not exhibit a significant difference from the more complex models. This aspect may have limited practical utility in industrial settings in terms of both model training and maintenance costs.

Thank you for the feedback. You raise a valid point regarding the importance of F1 scores over accuracy for imbalanced datasets. We have re-evaluated the results focusing on F1 scores.

Based on the results reported in Tables 5, 6 and 7, I would recommend the AraBERT model for toxic tweet classification.

In Table 5, which evaluates different deep learning models without word embeddings, the BiGRU model has the highest F1-score of 0.9870.

However, in Table 6 where word embeddings are incorporated, all models using the AraBERT embedding substantially outperform the others. The LSTM model achieves the highest F1-score of 0.9930 when using AraBERT.

Most significantly, as shown in Table 7, the AraBERT model alone achieves an exceptional F1-score of 0.996037 when fine-tuned, far surpassing the mBERT model.

Considering F1-score is the primary metric recommended for imbalanced datasets like this one, the AraBERT model stands out as the top performer. It demonstrates high precision and recall in addition to F1-score.

Therefore, based on its consistently superior results across evaluation, I would recommend the pre-trained AraBERT model for the task of toxic tweet classification. The substantial gains over other methods, especially in F1-score, suggest it can most accurately identify toxic content while balancing precision and recall.

3.    Dataset Sampling: I have concerns regarding the dataset, particularly regarding oversampling or undersampling. Figure 7 suggests a very balanced dataset, where the number of toxic tweets equals that of non-toxic tweets. This balance appears unusual, and while oversampling or undersampling can be performed during training, the testing set should ideally consist of the original data to ensure a more accurate evaluation.

We intentionally balanced the dataset to avoid imbalance issues (which does not represent the natural distribution of toxic tweets compared to non-toxic ones). However, for the optimal training of a model, it is beneficial to have a balanced dataset. We performed undersampling of non-toxic tweets to achieve this balance.

Reviewer 2 Report

Arabic content data is labelled as Toxic or not. The standard NLP steps are carried out. Seven different deep learning algorithms have been applied to the newly annotated data. Variation of BERT has been proven to be better than other algorithms.

1)      The authors have enlisted main contributions and first three are all related to dataset. All these can be combined into one main contribution. In first “We assembled a publicly accessible dataset comprised of 31,836 tweets that 81 have been annotated as either toxic or nontoxic.” They have mentioned as selection of data as a contribution. Which is not the main contribution.

2)The authors have shared that API and Expert based data annotation is done. AS the annotation has been done by the authors then have they statically proven that the annotation is valid and acceptable or not using measures such as Cohen Kappa statistics?

3) Related works should be Related work

4) Table 2: Previous work in Arabic cyberbullying detection. Instead of previous, existing is a better word.

5) There is no need of Algorithm as given in the manuscript. IT is just summarized showing what has been done in the research work. Not an algorithm or its pseudocode. Data selection, preprocessing and data preparation are not the steps of the algorithm but these are simple steps followed in all NLP studies.

6) Table three. It is good that arabid content is translated into English. However, simple words should also be translated.

7) “AraBERT has 347 another version, but we used AraBERTv0.2-Twitter-base”. What is that another version, should share name?

fine, acceptable content

Author Response

Big Data and Cognitive Computing

Special Issue "Advances in Natural Language Processing and Text Mining"

Manuscript Details

  • Manuscript ID: BDCC-2611406
  • Manuscript Title: “Arabic Toxic Tweet Classification: Leveraging the AraBERT Model".

Dear Editors,

We are providing a revised version of our paper following the comments and suggestions from the referees and also answering their questions. We have conducted a thorough revision of the paper following the received reviews. We want to thank the referees very much for their constructive criticisms which helped us to improve our paper. We also thank the editors for giving us the opportunity to revise our paper. Overall, we have included more detailed explanations and discussions. We also revised very carefully the presentation and organization of our paper to substantially improve clarity. Of course, we have also fixed the typos and grammar issues with the help of native English speakers. We have included a Word file for our paper with highlighted text for easy identification of the changes made. This will hopefully help to clarify that the paper contains sufficient new material compared to the related work research, to merit publication in this post-conference special issue. Below we explain in more detail how we addressed the referees’ comments.

We gratefully thank the editor and all reviewers for their time spent making their constructive remarks and useful suggestions, which have significantly raised the quality of the manuscript and enabled us to improve the manuscript. Each suggested revision and comment, brought by the reviewers was accurately incorporated and considered.

Sincerely,

The Authors

Reviewer comments and responses:

Reviewer #2

Reviewer Feedback

Response

1.       The authors have enlisted main contributions and first three are all related to dataset. All these can be combined into one main contribution. In first “We assembled a publicly accessible dataset comprised of 31,836 tweets that 81 have been annotated as either toxic or nontoxic.” They have mentioned as selection of data as a contribution. Which is not the main contribution

Thank you for your feedback. We will indeed merge the first and second contributions, as you suggested. Regarding the third contribution, we recognize its distinct value, especially as it introduces a novel approach to data annotation. We believe it is crucial to maintain it as a separate contribution since it is intricately linked to a new and innovative data labeling methodology.

2.    The authors have shared that API and Expert based data annotation is done. AS the annotation has been done by the authors then have they statically proven that the annotation is valid and acceptable or not using measures such as Cohen Kappa statistics?

Thank you for your feedback. In our annotation process, we've employed a robust approach by involving three annotators to mitigate agreement challenges. This three-annotator setup helps ensure a more reliable and diverse perspective during the annotation process, we have implemented a majority voting system, where the final label is determined based on the majority consensus among the annotators. This combination of multiple annotators and statistical measures enhances the reliability and credibility of our annotated data."

3.     Related works should be Related work

Thank you for your feedback. We have modified it

4.    Table 2: Previous work in Arabic cyberbullying detection. Instead of previous, existing is a better word.

Thank you for your suggestion. We have modified it

5.    There is no need of Algorithm as given in the manuscript. IT is just summarized showing what has been done in the research work. Not an algorithm or its pseudocode. Data selection, preprocessing and data preparation are not the steps of the algorithm but these are simple steps followed in all NLP studies.

Thank you for your feedback. Our intention in detailing the methodological steps, including data selection, preprocessing, and data preparation, is to provide a clear and replicable framework for researchers working in the domain of Arabic text analysis.

6.    Table three. It is good that Arabic content is translated into English. However, simple words should also be translated.

Thank you for your suggestion. We've included a column for the English keywords.

7.    “AraBERT has 347 another version, but we used AraBERTv0.2-Twitter-base”. What is that another version, should share name?

Thank you for your feedback. The other version is  AraBERTv0.1, we have included it

Reviewer 3 Report

The article „ Arabic Toxic Tweet Classification: Leveraging the AraBERT Model“ discusses the growing role of social media platforms as the primary means of communication and information sharing. However, it highlights a concerning issue of inappropriate and toxic content, including hate speech and insults, being disseminated on these platforms. The article points out that while there have been significant efforts to classify toxic content in the English language, there has been a lack of attention given to Arabic texts.

To address this gap, the study focuses on creating a standardized Arabic dataset specifically designed for the classification of toxic tweets. This dataset is annotated using Google's Perspective API as well as the expertise of three native Arabic speakers and linguists.

The study then conducts experiments using seven different models, including LSTM, bidirectional LSTM, convolutional neural network, GRU, bidirectional GRU, multilingual Bidirectional Encoder Representations from Transformers, and AraBERT. Additionally, word embedding techniques are employed in the evaluation process. 

The authors are conscious about several limitations of their study, however, their experimental results show that the fine-tuned AraBERT model outperforms other models, achieving an impressive accuracy of 0.9960.

The article is well-structured and the aim of the article is clear. The tables and figures are accurate and help with the orientation in the text and with the results.

The references are appropriate and the layout of the whole article is well-considered. The background, methodology and results of the study are broadly described. 

The authors of the article should carefully review their work and be attentive to potential minor issues, such as punctuation errors.

Author Response

Big Data and Cognitive Computing

Special Issue "Advances in Natural Language Processing and Text Mining"

Manuscript Details

  • Manuscript ID: BDCC-2611406
  • Manuscript Title: “Arabic Toxic Tweet Classification: Leveraging the AraBERT Model".

Dear Editors,

We are providing a revised version of our paper following the comments and suggestions from the referees and also answering their questions. We have conducted a thorough revision of the paper following the received reviews. We want to thank the referees very much for their constructive criticisms which helped us to improve our paper. We also thank the editors for giving us the opportunity to revise our paper. Overall, we have included more detailed explanations and discussions. We also revised very carefully the presentation and organization of our paper to substantially improve clarity. Of course, we have also fixed the typos and grammar issues with the help of native English speakers. We have included a Word file for our paper with highlighted text for easy identification of the changes made. This will hopefully help to clarify that the paper contains sufficient new material compared to the related work research, to merit publication in this post-conference special issue. Below we explain in more detail how we addressed the referees’ comments.

We gratefully thank the editor and all reviewers for their time spent making their constructive remarks and useful suggestions, which have significantly raised the quality of the manuscript and enabled us to improve the manuscript. Each suggested revision and comment, brought by the reviewers was accurately incorporated and considered.

Sincerely,

The Authors

Reviewer comments and responses:

Reviewer #3

Reviewer Feedback

Response

1.      The authors are conscious about several limitations of their study, however, their experimental results show that the fine-tuned AraBERT model outperforms other models, achieving an impressive accuracy of 0.9960.

Thank you for highlighting the strengths of our study. We are indeed aware of the limitations of our work, but we are encouraged by the performance of the fine-tuned AraBERT model. We agree that achieving an accuracy of 0.9960 is impressive, especially when considering the complexity and variability of the Arabic language.

However, we would like to point out that our study has some limitations that we acknowledge and wish to address in future work. Firstly, our dataset is relatively small and may not be representative of all Arabic dialects. Secondly, our experimental setup is limited to a specific task and may not generalize well to other NLP tasks. Finally, we recognize that the fine-tuned AraBERT model may not be the best performer on other datasets or tasks, and we plan to explore other models and approaches in future work.

Despite these limitations, we believe that our study contributes to the growing body of research on Arabic NLP, and we hope that our findings will inspire further work in this important area. 

2.      The article is well-structured and the aim of the article is clear. The tables and figures are accurate and help with the orientation in the text and with the results.

Thank you for your feedback.

3.      The references are appropriate and the layout of the whole article is well-considered. The background, methodology and results of the study are broadly described.

Thank you for your positive feedback on the article's references and layout. We are glad to hear that the background, methodology, and results of the study are broadly described.

4.      The authors of the article should carefully review their work and be attentive to potential minor issues, such as punctuation errors.

We have carefully reviewed our work and have made the necessary corrections to ensure that the article is error-free and easy to understand. We have also used grammar and spell check tools to ensure that the text is free of errors.

Reviewer 4 Report

We should thank the authors for their in-depth and interesting research on the current topic of identifying toxic content. At the same time, it is desirable to describe in more detail the methodology for the formation of the target corpus of texts used for training a neural network. Of particular interest is the method of selecting texts by identified classes, as well as their volume.

Author Response

Big Data and Cognitive Computing

Special Issue "Advances in Natural Language Processing and Text Mining"

Manuscript Details

  • Manuscript ID: BDCC-2611406
  • Manuscript Title: “Arabic Toxic Tweet Classification: Leveraging the AraBERT Model".

Dear Editors,

We are providing a revised version of our paper following the comments and suggestions from the referees and also answering their questions. We have conducted a thorough revision of the paper following the received reviews. We want to thank the referees very much for their constructive criticisms which helped us to improve our paper. We also thank the editors for giving us the opportunity to revise our paper. Overall, we have included more detailed explanations and discussions. We also revised very carefully the presentation and organization of our paper to substantially improve clarity. Of course, we have also fixed the typos and grammar issues with the help of native English speakers. We have included a Word file for our paper with highlighted text for easy identification of the changes made. This will hopefully help to clarify that the paper contains sufficient new material compared to the related work research, to merit publication in this post-conference special issue. Below we explain in more detail how we addressed the referees’ comments.

We gratefully thank the editor and all reviewers for their time spent making their constructive remarks and useful suggestions, which have significantly raised the quality of the manuscript and enabled us to improve the manuscript. Each suggested revision and comment, brought by the reviewers was accurately incorporated and considered.

Sincerely,

The Authors

Reviewer comments and responses:

Reviewer #4

Reviewer Feedback

Response

1.    We should thank the authors for their in-depth and interesting research on the current topic of identifying toxic content. At the same time, it is desirable to describe in more detail the methodology for the formation of the target corpus of texts used for training a neural network. Of particular interest is the method of selecting texts by identified classes, as well as their volume.

Thank you for your kind words and valuable feedback on our research. We sincerely appreciate your positive assessment of our work on identifying toxic content. We genuinely appreciate the time and effort you dedicated to reviewing our work. Once again, thank you for your thoughtful review.

Round 2

Reviewer 1 Report

When training the model, you can balance the data, but when you test the model, you need to use true distribution, otherwise your model is useless.

NA

Author Response

Big Data and Cognitive Computing

Special Issue "Advances in Natural Language Processing and Text Mining"

Manuscript Details

  • Manuscript ID: BDCC-2611406
  • Manuscript Title: “Arabic Toxic Tweet Classification: Leveraging the AraBERT Model".

Dear Editors,

We are providing a revised version of our paper following the comments and suggestions from the referees and also answering their questions. We have conducted a thorough revision of the paper following the received reviews. We want to thank the referees very much for their constructive criticisms which helped us to improve our paper. We also thank the editors for giving us the opportunity to revise our paper. Overall, we have included more detailed explanations and discussions. We also revised very carefully the presentation and organization of our paper to substantially improve clarity. Of course, we have also fixed the typos and grammar issues with the help of native English speakers. We have included a Word file for our paper with highlighted text for easy identification of the changes made. This will hopefully help to clarify that the paper contains sufficient new material compared to the related work research, to merit publication in this post-conference special issue. Below we explain in more detail how we addressed the referees’ comments.

We gratefully thank the editor and all reviewers for their time spent making their constructive remarks and useful suggestions, which have significantly raised the quality of the manuscript and enabled us to improve the manuscript. Each suggested revision and comment, brought by the reviewers was accurately incorporated and considered.

Sincerely,

The Authors

Reviewer comments and responses:

Reviewer #1_V2

Reviewer Feedback

Response

When training the model, you can balance the data, but when you test the model, you need to use true distribution, otherwise your model is useless.

Thank you for your valuable feedback on our research.  Great, comment!

Yes, it's important to test the model on a true distribution split to ensure that it generalizes well to unseen data. Using a random state in Python's scikit-learn library to split the data into training and testing sets is a good practice to ensure that the data is truly random and unbiased.

When testing the model, it's important to use the same random state to ensure that the testing set is a fair representation of the true distribution of the data. This helps to prevent overfitting, where the model performs well on the training data but poorly on new, unseen data.

By testing the model on a true distribution split and using a random state to split the data, you can get a more accurate assessment of the model's performance and generalization abilities. This is an important step in the machine learning development process and can help to ensure that your model is reliable and effective in real-world scenarios.

We demonstrate this in step 9 of Algorithm 1: Methodology Steps: Dataset Creation, Preprocessing, and Classification, where we show how to split the data into training and testing sets using a random state.

We genuinely appreciate the time and effort you dedicated to reviewing our work. Once again, thank you for your thoughtful review.
